# Consumer Expectation and Perception of Farmed Rainbow Trout (*Oncorhynchus mykiss*) Fed with Insect Meal (*Tenebrio molitor*)

**DOI:** 10.3390/foods12234356

**Published:** 2023-12-02

**Authors:** Martina Magnani, Anna Claret, Enric Gisbert, Luis Guerrero

**Affiliations:** 1Department of Veterinary Medical Sciences, University of Bologna, Via Tolara di Sopra 50, Ozzano Emilia, 40064 Bologna, Italy; martina.magnani20@unibo.it; 2Food Quality and Technology, Institut de Recerca i Tecnologia Agroalimentàries—IRTA, Finca Camps i Armet s/n, 17121 Monells, Spain; anna.claret@irta.cat; 3Aquaculture Program, Centre de la Ràpita, Institut de Recerca i Tecnologia Agroalimentàries—IRTA, Crta Poble Nou km 5.5, 43540 La Ràpita, Spain; enric.gisbert@irta.cat

**Keywords:** insect meal, *T. molitor*, sensory profile, consumer expectation, consumer acceptance, Rainbow trout, *Oncorhynchus mykiss*, willingness to buy, willingness to pay

## Abstract

In recent years, insect meal has attracted increasing interest as an innovative protein source to replace fish meal in feed formulations due to its valuable nutritional profile. This research aimed to compare the effects of different levels of dietary inclusion of the yellow mealworm beetle (*T. molitor)* larvae meal on the sensory quality of rainbow trout (*Oncorhynchus mykiss*) fillets and retrospectively on the acceptability of this protein source to consumers. The results showed that the inclusion of *T. molitor* larvae meal did not induce sensory changes in the trout fillets, while regarding consumer acceptability and willingness to buy and pay, it was shown that a certain level of rejection towards this alternative protein still exists. The work described in this scientific manuscript adds more knowledge on the study of consumer acceptability of this protein source.

## 1. Introduction

According to the latest data published by the Food and Agriculture Organization of the United Nations (FAO), world aquaculture production in 2020 was 122.5 million tonnes, an increase of 3% compared to the previous year and with an estimated average annual growth rate of around 3.3%. These figures can be explained by the increasing global demand for healthy and nutritious fish and fish products because of the growing world population and changes in eating habits (consumption patterns) [1]. The expected increase in consumption of fish products therefore suggests a necessary adoption of new and more sustainable aquaculture production systems.

There are several obstacles to the expansion of aquaculture, including the application of more restrictive environmental regulations, the decrease in available water and climate change [1]. It is precisely the latter that have a very direct impact on aquaculture [2], and although these changes vary between geographical areas, the Mediterranean region is expected to be affected faster and more acutely, in some respects, than other areas [3]. Aspects such as increasing temperature, changing salinity, decreasing pH, rising sea levels, and changes in ocean productivity, may directly or indirectly impact fisheries and aquaculture [4]. The link between these two activities is highly correlated, because it is through the capture of wild fish resources that fishmeal and fish oil can be produced and used in the production of feed for aquaculture [5]. From 2000 onwards, fishmeal (FM), for feed production, became a product of limited availability precisely due to the rapid expansion of the aquaculture sector and the consequent demand for this product outstripping supply [6].

The need, particularly for carnivorous species, for protein in the diet is rather high in both quality and quantity. For this reason, FM has until now been considered the best protein source for feed formulation [7]. Since a few years ago, the scarcity of this raw material has forced us to explore new alternative and sustainable protein sources [8].

Over the last decade, different alternative, plant [9,10] and non-plant sources of protein have been tested, and the growing demand for these new sources of protein and nutrients has led to the observation that, in many parts of the world, insects are already used as food for human and animal consumption. This has prompted scientists to investigate the nutritional properties of this alternative source. In particular, T. molitor meal is used in this study. This meal has a high protein value (50% for fat meal up to 82% for fat-free meal) compared to the protein content of good quality FM, which can reach up to 73%, and soya meal, which can contain up to 50% protein [11]. Insect meals, and in particular *T. molitor*, are very rich in essential amino acids, which makes them of high nutritional value. In addition, high levels of minerals such as iron, magnesium, zinc and selenium and low amounts, compared to FM, of calcium and potassium were observed. Vitamins are also contained in a good percentage [12]. The lipid content within *T. molitor* can be up to 30%, in contrast to the content found in FM (8.2%) and soybean meal (3%) [11].

The insect diet is mainly responsible for the variations in lipid and fatty acid (FA) composition [12]. In fact, the fat content is rich in long chain fatty acids such as oleic acid, linoleic acid, and palmitic acid. These unsaturated fatty acids (ω3) are present in plant products and are useful in lowering blood pressure and cholesterol levels in humans. Eicosapentenoic acid (EPA) and docosahexaenoic acid (DHA) normally found in animal products are also present, but in trace amounts. In addition, *T. molitor* fats are also rich in polyunsaturated fatty acids (ω6) such as linolenic acid, which is also normally found in plant products, while arachidonic acid (ARA), which is found in animal products, is only present in trace amounts. Polyunsaturated fatty acids are involved in the production of the lipid component of all cell membranes, both animal and human [13].

Nutritional analyses on the meal derived from this insect type meant that it could be used and evaluated as an ingredient in diets for livestock [14] and for feeding in aquaculture species. In these studies, it has been shown that it is possible to replace FM with *T. molitor* meal at doses of no more than 50%; in fact, above this percentage, a reduction in growth rate and fattening was noted [8,15,16,17,18,19]. These studies corroborated to show that *T. molitor* meal is a viable alternative to the use of fish ingredients in animal feed.

Given the growing interest in insect products as an alternative source of protein, the acceptability and knowledge of consumer preferences for insect foods may be critical. This knowledge would enable informed decisions on the production of these foods, particularly for Western countries where insect food consumption is largely non-existent [20]. These studies are, to date, scarce [21]; however, we know that humans generally avoid unfamiliar foods because they suffer from neophobia [22]. Although we acknowledge the contribution of such studies, we agree with other authors who argue that insects could be more easily introduced into consumers’ daily diets by developing products that involve eating insects indirectly such as using insect-based feed [23,24]. This was also demonstrated by Verbeke et al. [25] who in research conducted in Belgium, concluded that consumers are more willing to accept insects as animal feed than consume them directly in their diets. In the same vein, La Barbera et al. [26] pointed out that acceptance of indirect entomophagy does not necessarily indicate acceptance of direct entomophagy. In fact, for most Western consumers insect food seems to be good for animals and not very appropriate for humans [27]. However, it is difficult to generalize, as there may be segments or groups of consumers who are more open to exploring new flavours and new products. For instance, studies such those conducted by Baldi et al. [28] and Sogari et al. [29] point out that younger consumers, regardless of gender, occupation or education, but with a slightly higher income, trust innovation in food production and are active in the search for new food sources.

This study is a step towards an even greater understanding of consumer acceptance of new products, such as fish raised on insect-based feed. Thus, the main objective of this study was to explore the perception and expectations of Spanish consumers of trout fed with different amounts of insect meal.

## 2. Materials and Methods

### 2.1. Ethical Approval

The study was approved by the Ethics Committee of the Institute of Agri-Food Research and Technology (IRTA), registration number CCSC 31/2023, in accordance with legal requirements related to ethical principles of research with human participants (Declaration of Helsinki and Belmont Report). Each participant provided written informed consent to take part in the study.

### 2.2. Fish Sample

Rainbow trout (*Oncorhynchus mykiss*) were fed with feed four isoproteic (42%), isolipidic (25%) and isoenergetic (22.6 MJ/kg feed) diets containing graded levels of defatted insect meal from *T. molitor* instead of fish meal: control (CRL 0%), 30%, 60%, 100% insect meal supplied by TEBRIO (Doñinos de Salamanca, Salamanca, Spain). Fish were fed the abovementioned experimental diets for 60 days by triplicate at IRTA research facilities in la Ràpita (Tarragona, Spain) under standard rearing conditions (feeding rate 1.5%, water temperature of 16–17 °C). At the end of the fattening phase, no differences in fish body weight were found among dietary groups, and the average body weight ranged between 468.3 to 485.9 g. Fish were sacrificed in ice slurry and then, they were filleted and packed in plastic bags and frozen. The fillets were then transported to the IRTA in Monells where they were again vacuum-packed in aluminium food bags to prevent oxidation and stored at −18° until analysis.

### 2.3. Sensory Analysis

This study was carried out by a sensory panel of seven individuals, all of whom had previous experience in sensory analysis of different types of products including fish and fish products. The panellists were asked to identify and generate sensory attributes to characterise the products that were fed these experimental diets. The 23 descriptors shown in Table 1 were chosen by means of the Check-All-That-Apply (CATA) method from a list of 58 attributes obtained from those previously generated by the tasters and a literature work [30,31,32,33,34]. This process was performed in two tasting sessions involving the seven available tasters.

Once the final list of descriptors was defined, to facilitate the homogeneous use of all of them by the panellists [35], reference scales were used following those proposed by Lazo et al. [36] to define high and low values for each selected sensory descriptor. Once the final form was assembled, the samples were evaluated quantitatively in three different sessions using the quantitative descriptive analysis method [37,38].

Samples that had been stored for 5 months in cold storage at −18° were thawed the evening before the analysis by placing them in the refrigerator at 4°. The following morning the fillets were cut into two pieces from each fillet, using only the dorsal side, and placed in individual transparent glass jars (model B-250, Juvasa, Spain) to make the sample viewable for appearance analysis.

The samples were then baked in a conventional oven at a temperature of 115° for 14 min. The lids of the jars were used throughout the preparation and baking until the final evaluation to preserve the smell of the samples. Immediately after cooking, the jars were placed inside a portable electric heater (Solac, model 212, 220–240 V, Taurusgroup, Oliana, Spain) to keep them warm until tasting. In each session, the order in which the samples were presented and the first order and carry over effects were balanced using an appropriate experimental design [39].

The sensory evaluation was performed in a tasting room designed according to ISO guidelines [40]. Each panellist evaluated the samples placed inside the portable heating device in a set order. The samples were evaluated using a semi-structured 10 cm linear scale anchored at both ends (0 when the descriptor was absent and 10 for maximum expected intensity in this product category). All panellists used water, to be drunk between samples, as a palate neutraliser.

### 2.4. Consumer Acceptance

#### 2.4.1. Participants

A total of 116 consumers were recruited between the cities of Barcelona (56 consumers) and Madrid (60 consumers) through the marketing agency Silliker S.A.U. (Mérieux NutriSciences, Barcelona, Spain) using a questionnaire specifically created for this study. Participants had to be consumers of the species under study, be between 18–70 years of age and comprise 50% women and 50% men, be willing to taste fish fillets fed with feed that included meal of animal origin (crustacean, insect, egg protein) instead of fish meal.

#### 2.4.2. Questionnaire

To assess consumers’ expectations and behaviour towards insect-fed animal products, we designed and distributed a questionnaire divided into seven parts in which blind and informed tasting, expectations, perception and willingness to buy and pay for different products were assessed.

Consumers tasted the samples and filled in the corresponding questionnaire in one day per city, in sessions with 20 participants at a time.

At the beginning of each session a short presentation of the project was made which explained how the work to follow would be organised.

After signing the informed consent, they were presented with the first part of the questionnaire, which consisted of scoring the liking of the different products blindly after tasting them. A 9-point hedonic scale ranging from 1 = dislike extremely to 9 = like extremely was used [41].

The second part of the questionnaire asked about their expectation, i.e., how much they thought they would like the insect meal-fed product, again using a 9-point liking/disliking scale (1 = expect to “dislike extremely”, 9 = expect to “like extremely”) [42]. The same question was also proposed to consumers for the use of crustacean meal and egg protein to compare consumer acceptance of more familiar alternative proteins to insect meal, and so that they would not know until then that our focus was on the use of insects. 

The third part of the questionnaire again involved scoring the liking of the product, this time the tasting was informed and the consumer knew what percentage of insect meal had been fed into the product they tasted or whether it was the control sample. This procedure was performed to assess the rejection of the use of this meal even when used as feed [43]. The evaluation of liking was carried out in the same way as for the blind sample.

The fourth part involved measuring consumers’ perceptions of trout fed with insect meal [21,44,45,46,47]. The scale used consisted of seven items, but the questions asked were divided into two sections. One of these was asked immediately after the informed tasting of the products using the positive and negative term of the parameter to be scored as the extremes of the scale. The questions asked were: Known–Unfamiliar, Innovative–Common, Safe–Unsafe, Healthy–Unhealthy, Expensive–Cheap, Bad Taste–Good Taste, Low Quality–High Quality, Sustainable–Non-Sustainable, Artificial–Natural, Environmentally Harmful–Environmentally Respectful.

The other section was asked at a later stage when the whole tasting part was finished. In this part, the scale available for scoring had as extremes: 1 = totally disagree, 7 = totally agree and it was asked whether trout fed with insect meal would be nutritious, would be healthy, would taste good, would be expensive, would be difficult to digest, were produced in an environmentally friendly way, would be of high quality, would be safe for health, would be more sustainable.

The fifth part was focused on assessing the willingness to buy the different products considering that they have a normal price. The scale used was an 11-point scale, and the minimum value had “there is no or almost no chance that I would buy it” (1% chance) and as the maximum “it is certain or practically certain that I will buy it” (99% chance) [48,49].

The sixth part asked how much they would be willing to pay for this product. The scale had as a central value 0% which means that they would be willing to pay the same as they currently pay, negative percentages that they would pay less and positive percentages that they would be willing to pay more for this product [20,50].

The last, and seventh part, collected sociodemographic information such as gender, age, education, economic situation, frequency of trout consumption and whether they were responsible for buying the groceries at home.

### 2.5. Statistical Analysis

Differences between dietary treatments were tested by a three-way ANOVA with a significance level of 95% (*p* ≤ 0.05), including dietary treatments and assessors as fixed factors and testing session as a random factor. Tukey’s post hoc significant difference test was applied for comparing the mean values of the different treatments. 

For the statistical analysis of the consumer data set, a three-way ANOVA was used in which the consumer was placed as the random factor and the dietary treatments and cities as the fixed factor. This was followed by a Tukey’s post hoc significant difference test for comparing the mean values of the different treatments. 

The data were analysed using the XLSTAT statistical software, Version 19.6 (2020) (Addinsoft, Paris, France).

## 3. Results and Discussion

### 3.1. Sensory Analysis

Changes in fish feed ingredients can affect the appearance, smell, and aroma of fish fillets [51]. This could in turn, affect the perceived quality of the fillet and consequently consumer acceptance [32]. Indeed, Borgogno et al. [33], measuring sensory attributes and physicochemical parameters, found differences in the perceived intensity of odour, flavour and texture in rainbow trout fillets fed a diet that included insect meal. These results disagree with previous findings reported in the literature. For example, no significant sensory differences were found with the inclusion of insect meal in diets for Atlantic salmon [31]. In the same vein, Sealey et al. [52] found no significant differences when performing a triangle test on rainbow trout fed diets with different insect meal content.

The sensory analysis performed in this study on the intensity data of sensory attributes to estimate the sample effect (CRL 0%, 30%, 60%, 100%) (Figure 1) found no significant differences (*p* ≤ 0.05) in any case, in agreement with Iaconisi et al. [53]. The lack of statistical differences in this study is probably related to the presence of other notably intense attributes such as earthiness, which could have masked the possible effect of diet on sensory characteristics as observed by Lazo et al. [36] in pikeperch (*Sander lucioperca*).

There were no significant flavour differences between the diet groups, but the numerical scores for earthy, sour, bitter and for flavour intensity and persistence tend to increase with the maximum level of inclusion of *T. molitor* meal, as observed in other studies [33].

### 3.2. Consumer Acceptance and Perception

In the face of increasing global threats to food security, public health, and environmental sustainability, insects are considered a new source of human food and animal feed in Western countries [54,55]. However, despite the advantages of insects as a sustainable and healthy alternative to conventional protein sources, they are generally not considered a viable food source by Western consumers [20]. The results show that consumers are more willing to accept products that fit into their traditional diet and that the acceptance of insects as human food also depends greatly on the composition and level of feed processing [56]. 

To assess consumers’ expectation of products fed with insect meal, in this study we asked the question: how much they think they might like a product fed with insect meal, a foodstuff not part of the human diet in Western countries, compared to a product fed with crustacean meal and egg protein, two products, therefore, commonly consumed in the West. As noted by Tan et al. [48], the presence of an invisible insect does not necessarily improve the overall liking of the entire insect product. Even motivated consumers still hesitate to regularly consume insect-based foods due to other practical and socio-cultural barriers. Our results are in line with the findings of Tan et al. [48], and Onwezen et al. [56] said, the expectation towards insect meal was punctuated with a lower value compared to crustacean and egg protein (Figure 2). The decrease in expectations indicates a worse global perception of insect meal, most likely because it is a product with which participants are unfamiliar compared to crustaceans or eggs. In any case, it should be noted that the mean value obtained was above the central point of the scale (5), indicating that it was not rejected.

Furthermore, when there is no real experience or regular consumption of a food, consumer expectations tend to be based on visual appearance and expected taste. Rozin and Fallon [57] showed that considering insects as food can evoke disgust and the thought of consumption can lead to the expectation and perception of bad taste. 

Table 2 shows that, for the known–unknown attribute, CRL was statistically different between 30, 60 and 100% insect meal, as might be expected, given that fish meal is commonly better known as an ingredient for an animal feed.

Another interesting and statistically significant result is found in relation to the healthy term. In this case and despite the score being always higher than the central point of the scale (4), the highest value was observed for the 0% CRL product and the lowest for the product with 100% insect meal. Based on this result, it seems that insect meal might still be perceived by consumers as a product potentially unsafe for human health because it can carry possible diseases and allergies. Given the nutritional value of insect meal, one way to increase consumer acceptance may be to inform them about the healthy properties of this type of diet [58]. Furthermore, according to Sogari et al. [21] it also has a positive impact on animal health by increasing digestive performance. As mentioned above, Rozin and Fallon [57], demonstrated that considering insects as food can evoke disgust, that can also be related to bad taste. Accordingly, the statistically significant result for the good taste term shows a higher score for the CRL 0% product that gradually decreases as the amount of insect meal increases. However, this contrasts with the sensory data we presented, which show no statistically significant differences in the sensory properties (Figure 1). Regarding the environmentally friendly term, the results show a statistically significant lower value for the CRL 0% product due to the use of fishmeal, compared to the 100% product. Therefore, it seems that participants were aware of the impact of using fish meal in fish diets. These data are in line with those shown by Benito et al. [3] in which it is predicted that the Mediterranean region may experience, in some respects, faster and more acute climatic changes than other areas and thus impact directly or indirectly on fisheries and aquaculture with an ever-decreasing availability of fishmeal to be used as a feed ingredient.

As shown by the results obtained for the overall perception (Table 3), the fish fed with insect meal is perceived by consumers as not being of high quality (4.683), contrary to what Sogari et al. [21] says. However, the insect meal-fed product is perceived by consumers positively when looking at the terms Nutritious (5.611), Natural (5.5.91), and Sustainable (5.5419). It is important to note that these results refer to the evaluation of the incorporation of insect meal to the fish diet without indicating the percentage of substitution, i.e., in a broad way and evaluated by means of a 7-point Likert scale.

To assess how information influences the acceptability of the product, participants, first tasted the samples blindly and then in an informed condition, thus scoring liking on a hedonic 9-point scale [41]. According to the results of the informed test, liking tended to decrease as the amount of insect meal within the product increased, probably because they were conditioned by the knowledge of the presence of this product within the sample they were tasting. This effect, in fact, was not observed in the blind tasting (Table 4). The lower acceptability expectation also observed for this type of diet compared to other alternative proteins to fishmeal (Figure 2), could also explain the decrease in acceptability in the informed tasting.

The commercial potential of products containing unprocessed and processed insects will depend heavily on the sensory liking, both before and after tasting [59]. The latest food technologies are a valuable aid to alleviate disgust and even reduce the impact of neophobia, even if affective-emotional reactions are only partially modified by awareness and information [26].

Regarding the willingness to buy, as could be expected, the results show the CRL 0% product with the highest value (7.345) and gradually this value decreases as the inclusion of insect meal increases (Table 5). These figures show how strong the rejection towards this foodstuff still is, even though the question was asked after the informed tasting. Regarding the willingness to pay for these products, our results show that the consumer is willing to pay a higher price for the CRL 0% product and a lower price as the insect meal substitution increases. This is not in line with Deely et al. [60], according to which, consumers are willing to pay more for environmentally friendly food products. Furthermore, it was noted that the willingness to pay more for such a product is positively correlated with the economic situation of the consumer.

## 4. Conclusions

This study contributes to the scientific literature by examining consumer expectations, perception and acceptance of an innovative product, insect-fed fish. The focus of the work is also of great interest in the context of the “sustainability”, which sees insect-fed fish as a possible future reality. Furthermore, considering that a limitation of this study is the specific national/cultural context (Spanish consumers), future experiments will have to make a cross-cultural comparison. Future research should also consider other consumption contexts (restaurant or home consumption), the stage of preparation (raw or ready-to-eat) and other fish species, with less pronounced sensory attributes (earthy flavour in the case of trout). Further studies are also needed to better identify different consumer groups or segments, which did not make sense in the present study given the relatively small number of participants. Practitioners should therefore try to publicise not only the potential benefits of entomophagy, but also consider taste education as an important tool to change attitudes and negative expectations towards edible insects [59]. The positive experience of tasting products with both visible and processed insects may cause consumers to reconsider their initial negative expectations and attitudes towards entomophagy, encouraging others to eat these novel foods as well. Furthermore, it would be interesting to investigate whether approval from others (e.g., family and friends) may be one of the most important factors for the introduction and spread of entomophagy as already explained in other papers [61,62].

## Figures and Tables

**Figure 1 foods-12-04356-f001:**
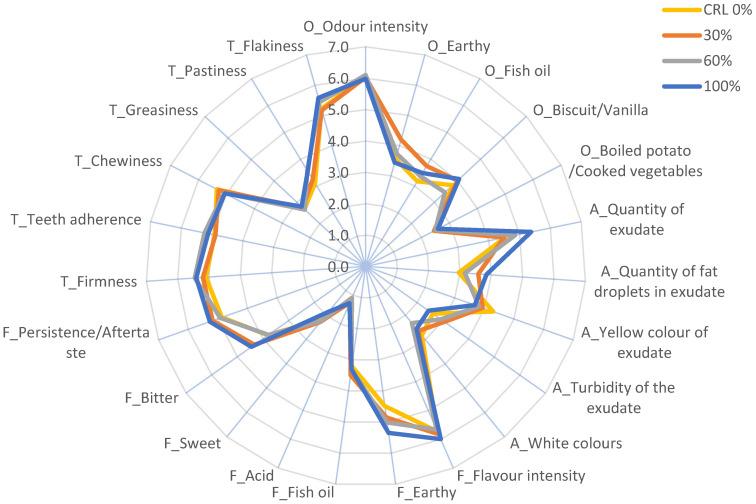
Sensory profile (mean values) of trout (*Oncorhynchus mykiss*) fillets fed four different diets (CRL 0%, 30%, 60%, 100% of insect meal) obtained with the trained tasters. Intensity values were obtained in a 10 cm unstructured line scale, anchored, respectively, at 0 (absence) and 10.0 cm (high intensity) from the beginning of the scale itself. No significant differences between treatments were observed (*p* > 0.05).

**Figure 2 foods-12-04356-f002:**
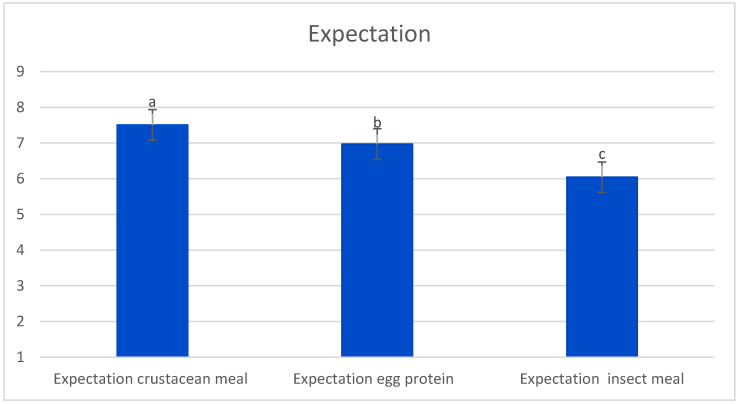
Histogram diagram of consumer expectation of insect meal in comparison with crustacean meal and egg protein (1 = expect to dislike extremely, 9 = expect to like extremely). Columns with different letters indicate significant differences between them (*p* ≤ 0.05).

**Table 1 foods-12-04356-t001:** Selected descriptors used for the final descriptive profile along with their description.

Attributes	Description
ODOUR	
Odour intensity	Intensity of odour
Earthy	Intensity of odour like humid earth
Fish oil	Intensity of characteristic odour
Biscuit/Vanilla	Intensity of odour like biscuits
Boiled potato /Vegetables	Intensity of odour like cooked vegetables
APPEARENCE	
Exudate quantity	Quantity of liquid released after cooking the sample
Fat droplets	Fat released in fish exudate in the form of oil droplets
Colour intensity of exudate	Yellow colour intensity of the exudate
Turbidity exudate	Suspended particles in exudate that block transparency
Colour intensity	Colour intensity from white to light brown inside the flesh of the fish
FLAVOUR	
Flavour intensity	Intensity of flavour
Earthy	Intensity of flavour like humid earth
Fish oil	Intensity of characteristic flavour
Acid	Intensity of flavour like citric acid
Sweet	Intensity of flavour like sugar
Bitter	Intensity of flavour like caffeine
Persistence/Aftertaste	Duration of stimulus in the oral cavity after swallowing
TEXTURE	
Firmness	Force required to deform the fillet between the tongue and palate
Teeth adherence	Degree in which fish sticks between molars
Chewiness	Number of chews before swallowing
Greasiness	Surface attribute that expresses the perception of the quantity of fat present in a product.
Pastiness	Degree in which fish turns into a paste after chewing
Flakiness	Degree of fish disintegration in the first bite

**Table 2 foods-12-04356-t002:** Mean value of consumer perception terms of the semantic differential scale (different letters in the same row indicate statistically significant differences (*p* ≤ 0.05)).

Pair of Adjectives	CRL 0%	30%	60%	100%	*p* Value
Known–unknown	2.845 ^b^	4.928 ^a^	4.879 ^a^	5.224 ^a^	<0.0001
New–common	4.897 ^a^	2.750 ^b^	2.560 ^b^	2.603 ^b^	<0.0001
Safe–unsafe	2.422 ^a^	2.897 ^b^	2.698 ^b^	2.828 ^b^	0.002
Unhealthy–healthy	5.793 ^a^	5.448 ^b^	5.401 ^b^	5.241 ^b^	<0.0001
Expensive–cheap	4.026	4.078	4.121	4.207	0.456
Bad taste–good taste	5.431 ^a^	5.103 ^ab^	4.984 ^b^	4.914 ^b^	0.002
Low quality–high quality	5.026	4.845	4.836	4.828	0.269
Sustainable–unsustainable	3.293	3.026	3.083	2.966	0.222
Artificial–natural	5.207 ^a^	4.853 ^ab^	4.948 ^b^	4.991 ^b^	0.049
Environ. unfriendly–environ. friendly	5.009 ^b^	5.267 ^ab^	5.397 ^a^	5.388 ^a^	0.004

**Table 3 foods-12-04356-t003:** Overall perceptions of products fed with insect meal (7-point Likert scale).

Overall Perception	Mean Score
Percep. Nutritious	5.611
Percep. Healthy	5.341
Percep. Good taste	4.864
Percep. Natural	5.091
Percep. Expensive	4.282
Percep. Difficulty digesting	4.028
Percep. Environ. responsible	5.359
Percep. High quality	4.683
Percep. Safe	5.241
Percep. Sustainable	5.541

**Table 4 foods-12-04356-t004:** Table of mean values of blind and informed liking (1 = dislike extremely, 9 = like extremely) for each tasted diet.

Experimental Diets	Blind	Informed
Crl 0%	6.871	6.871
30%	6.836	6.603
60%	6.750	6.509
100%	6.862	6.681
*p* value	0.856	0.099

**Table 5 foods-12-04356-t005:** Table of mean value of willingness to buy and willingness to pay. Different letters in the same column indicate statistically significant differences (*p* ≤ 0.0.5).

	Willingness to Buy	Willingness to Pay
Crl 0%	7.345 ^a^	6.774 ^a^
30%	6.578 ^b^	6.289 ^b^
60%	6.491 ^b^	6.183 ^b^
100%	6.328 ^b^	6.070 ^b^
*p* value	<0.0001	0.001

## Data Availability

The data used to support the findings of this study can be made available by the corresponding author upon request.

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
