# Peer review of "Consumer Expectation and Perception of Farmed Rainbow Trout (Oncorhynchus mykiss) Fed with Insect Meal (Tenebrio molitor)"

_foods, 2023, doi:10.3390/foods12234356_

Round 1
Reviewer 1 Report
Comments and Suggestions for Authors
I thought this was an excellent manuscript and a well-designed study.
Admittedly, most of my comments are focussed on research design/methodology/analysis/reporting, but that is due more to my interests and what I tend to focus on, than their relative weakness.
185 - Correct "Expensive-Expensive"
229 - It is not clear that the spider graph (and relevant discussion) is from the blind tasting.
229 - From the method section, it seems that you did not ask the same sensory questions after the informed tasting. If you did, have you considered displaying the spider graphs side-by-side?
229 - Spider graph - some characteristics (O_Odour..., O_Flavour...) are unclear or indistinguishable from others (A_Quantity of..., A_Quantity of...) after being shortened
253 - Consider using paired-sample t-tests to see whether the expectation responses differed across the protein sources.
256 - Figure 2 would benefit from expanding the legend to include the upper extreme value (9) and the anchor labels (9= like extreme, 1=dislike extreme).
268 - Consider adding (0%) to CRT, and % to the other numbers. Also, a column heading of "Percentage of Insect meal" here and in Figure 1 and Table 4 would increase clarity.
271 - Correct "Unknow"
295 - It is unclear whether Table 3 (and the relevant discussion) is reporting the result from the semantic differential scales or the Likert scales in Part 4 of the methodology discussion.
310 - It is unclear whether the numbers report the group means of a single item measure or a multi-item scale. This could be best clarified in the methodology section.
Finally, since you are reporting means and you are using a variety of response scales (9-post hedonic scales, 7-point semantic differential or variant scales, and 7-point Likert scales, consider adding a note to the Tables/Figures that indicate the anchor points of those scales.
Comments on the Quality of English Language
Language was very good and easy to read. I found some minor language errors that I included in my suggestions for authors.
Author Response
Consumer expectation and perception of aquaculture trout (Oncorhynchus mykiss) fed with insect meal (Tenebrio molitor)
foods-2720890
Review report 1
185 - Correct "Expensive-Expensive"
Corrected
229 - It is not clear that the spider graph (and relevant discussion) is from the blind tasting.
As explained in Materials and Methods, the sensory profile was obtained using the Quantitative descriptive Analysis (QDAÒ) method. This type of analysis requires trained panelists to evaluate the samples and them were coded with three-digit numbers so that they cannot know each sample to which treatment it corresponds. Thus, it was a blind tasting but with trained assessors.
229 - From the method section, it seems that you did not ask the same sensory questions after the informed tasting. If you did, have you considered displaying the spider graphs side-by-side?
The questions asked were different in each part except acceptability which was asked in both cases. After the blind tasting, only the acceptability was asked. In this case nothing else could be asked so as not to inform the consumers that the fish had been fed with insect meal. However, after the informed tasting (they already knew that the fish had been fed with insect meal) they were asked about their opinion and perception of it.
The spider graph refers to the results of trained tasters and not to consumers. Thus, from the blind tasting, expectations, and informed tasting with consumers, no such graph is available since they did not make a description of the sensory characteristics of the product as did the trained tasters.
229 - Spider graph - some characteristics (O_Odour..., O_Flavour...) are unclear or indistinguishable from others (A_Quantity of..., A_Quantity of...) after being shortened
Corrected
253 - Consider using paired-sample t-tests to see whether the expectation responses differed across the protein sources.
As indicated in the statistical analysis of material and methods section, a 3-factor analysis of variance was performed (treatment and city as fixed effects and consumer as a random effect). However, the graph was missing the letters indicating who was different from whom. That information has been added in this new version of the paper. Thank you for the comment.
256 - Figure 2 would benefit from expanding the legend to include the upper extreme value (9) and the anchor labels (9= like extreme, 1=dislike extreme).
Corrected
268 - Consider adding (0%) to CRT, and % to the other numbers. Also, a column heading of "Percentage of Insect meal" here and in Figure 1 and Table 4 would increase clarity.
Corrected
271 - Correct "Unknow"
Corrected
295 - It is unclear whether Table 3 (and the relevant discussion) is reporting the result from the semantic differential scales or the Likert scales in Part 4 of the methodology discussion.
Thank you for your comment, we fully agree that it was not clear neither in the table nor in the text. For this reason, we have added the corresponding clarification in both cases. These results refer to the global perception of the diet with insects and therefore correspond to the Likert scale.
Table 2 has also been transposed to better show whether the results are those of the semantic differential scale or those from the likert scale (semantic differential scale in this case).
310 - It is unclear whether the numbers report the group means of a single item measure or a multi-item scale. This could be best clarified in the methodology section.
Thank you again for your comment. The values shown refer to acceptability and therefore only include one item for both blind and informed assessment. This information has been added to table 4 to make it clearer.
Finally, since you are reporting means and you are using a variety of response scales (9-post hedonic scales, 7-point semantic differential or variant scales, and 7-point Likert scales, consider adding a note to the Tables/Figures that indicate the anchor points of those scales.
Corrected.

Reviewer 2 Report
Comments and Suggestions for Authors
The article deals with a very interesting topic, but:
- the final part of the introduction needs to be better focused
- the tables and figures showing the results are not well described and do not report important data such as the scales used or errors. The text that explains the results is very often unclear or not very incisive.
In the conclusions the authors talk about circular economy, but this aspect is not really explained throughout the article.
For detailed comments, see the attached file.

Comments on the Quality of English Language
Some sentences are too long and convoluted.
Author Response
Consumer expectation and perception of aquaculture trout (Oncorhynchus mykiss) fed with insect meal (Tenebrio molitor)
foods-2720890
Review report 2
Lines 52-55: Too long sentence
Corrected
Line 64: Please replace the comma with a period. Too long sentence.
Corrected
Reference n.22: You can find references about neophobia that are much more up to date than this one from 2005
Corrected with another reference published in 2023.
85-97 Too long and convoluted sentences. It is important to clarify these sentences to highlight the scope and purpose of your study
This paragraph has been modified to make it clearer and easier to read and to highlight the aim of the study.
How many consumers from Barcelona and from Madrid?
Corrected
Line 185; expensive-expensive? Correct it
Corrected
203- Why did you choose to ask only at the end of the questionnaire if the interviewees are regular consumers of trout?
Thanks for the question. The ideal situation would have been to select trout consumers, but that would make the recruitment process much more difficult. For that reason, we decided to limit ourselves to recruiting only fish consumers who were willing to try animals fed with animal origin diets different from fish. We also did not want to inform the participants in advance about what kind of species they would have to taste to avoid possible biases in the blind tasting. In any case, we considered that it was important to know if they were habitual trout consumers to explore if familiarity with the product could have any effect on their assessment (whether those with a higher consumption of trout could better detect possible modifications caused by the addition of insect flour). We did not observe any difference in this regard, so this aspect has not been included or discussed in the paper.
Lines 221-238: Figure 1 does not show whether there are statistical differences between the samples (P<=0.05): there are no error bars in the graph. You should change these sentences or add a table with values +- SD.
The non-existence of statistical differences between treatments for any attribute is mentioned in the text. This information has also been added in Figure 1 to make it clearer. The purpose of that graph is to provide a sensory map of the product, so it usually does not contain error bars. It is the standard way to present the results of sensory analysis and we prefer that format to a table, especially considering the absence of differences. However, if the editor deems it appropriate, we can replace the graph with a table.
Figure 1: Please explain better. This figure shows consumers' ratings of selected sensory attributes using the scale from 0 (corresponding to....) to 9 (corresponding to...).
As explained in the text (line 239 in the new submitted version), there are no statistical differences in the sensory attributes between treatments. This graph, commonly used in descriptive analysis (QDAÒ), shows sensory profiles obtained with trained tasters (not with consumers). The description of the scale used has also been included in this figure.
This sentence is useless. You said that there are no statistical differences!!!
Corrected
Not clear. Please rewrite it avoiding too long and convoluted sentences
Corrected, the paragraph has been deleted
Figure 2: error bars??? Moreover, please explain in the caption the scale you used.
Corrected
Lines 264-265: Table 2 shows that, for the Unknown, CRL is statistically different from 30, 60 and 100% insect meal
Table 2: please explain in the caption the value scales you used. For the columns "unknown", "unsafe", Unsustainable" (negative item) what do the highest values refer to? And for the positive items? You don't explain the questions you made to the consumers and how you report the results.
You can change the first row, indicating for each item the opposite terms, i.e known-unfamiliar, safe-unsafe...
Corrected. The table has been transposed to include the two adjectives used in each scale.
Lines 270-294: Results are poorly illustrated in the text. Not clear
Corrected, the paragraph has been modified accordingly.
Lines 306-309: not clear
Corrected
Line 332:Why circular economy? Throughout the article you don't explain why!
Corrected in the text.
The concept referred to was that of the sustainability of these protein sources, within feeds, as opposed to fish meal.
Line 345: it appears from the literature, but you didn't analyze this aspect. You should change the sentence with (for example): it would be interesting to verify if indirect approval from others (e.g. family and friends) may be one...
Corrected
Some references lack information
Corrected

Reviewer 3 Report
Comments and Suggestions for Authors
The paper entitled ‘Consumer expectation and perception of aquaculture trout (Oncorhynchus mykiss) fed with insect meal (Tenebrio molitor)’ investigates compare the effects of different levels of dietary inclusion of Tenebrio molitor larvae meal on the sensory quality of trout (Oncorhynchus mykiss) fillets and retrospectively on the acceptability of this protein source to consumers. I have no specific scientific concern but it is mandatory to correct the manuscript in some points:
1. As Fig. 1 shown, the numerical scores for earthy, sour, bitter and for flavour intensity and persistence increased with the maximum level of inclusion of Tenebrio molitor flour. Please try to explain.
2. As Fig. 2 shown, expectation towards insect meal was punctuated with a lower value compared to crustacean and egg protein. Please try to explain factors.
Author Response
Consumer expectation and perception of aquaculture trout (Oncorhynchus mykiss) fed with insect meal (Tenebrio molitor)
foods-2720890
Review report 3
As Fig. 1 shown, the numerical scores for earthy, sour, bitter and for flavour intensity and persistence increased with the maximum level of inclusion of Tenebrio molitor flour. Please try to explain.
We are sorry, but we do not have enough information to be able to justify this tendency in certain flavors. It is most likely a consequence of the addition of insect meal, but we cannot say for sure.
As Fig. 2 shown, expectation towards insect meal was punctuated with a lower value compared to crustacean and egg protein. Please try to explain factors.
This aspect is now better explained in the text, in the paragraph: Consumer acceptance and perception.
